# Analysis of *bla*_CHDL_ Genes and Insertion Sequences Related to Carbapenem Resistance in *Acinetobacter baumannii* Clinical Strains Isolated in Warsaw, Poland

**DOI:** 10.3390/ijms22052486

**Published:** 2021-03-02

**Authors:** Alicja Słoczyńska, Matthew E. Wand, Stefan Tyski, Agnieszka E. Laudy

**Affiliations:** 1Department of Pharmaceutical Microbiology, Medical University of Warsaw, PL 02-097 Warsaw, Poland; alicja.sloczynska@gmail.com (A.S.); s.tyski@nil.gov.pl (S.T.); 2Public Health England, National Infection Service, Porton Down, Salisbury SP4 0JG, UK; Matthew.Wand@phe.gov.uk; 3Department of Antibiotics and Microbiology, National Medicines Institute, PL 00-725 Warsaw, Poland

**Keywords:** multidrug resistance, carbapenem resistance, CHDL enzymes, insertion sequences, CarbAcineto NP test, PFGE, MLST

## Abstract

*Acinetobacter baumannii* is an important cause of nosocomial infections worldwide. The elucidation of the carbapenem resistance mechanisms of hospital strains is necessary for the effective treatment and prevention of resistance gene transmission. The main mechanism of carbapenem resistance in *A. baumannii* is carbapenemases, whose expressions are affected by the presence of insertion sequences (ISs) upstream of *bla*_CHDL_ genes. In this study, 61 imipenem-nonsusceptible *A. baumannii* isolates were characterized using phenotypic (drug-susceptibility profile using CarbaAcineto NP) and molecular methods. Pulsed field gel electrophoresis (PFGE) and multi-locus sequence typing (MLST) methods were utilized for the genotyping. The majority of isolates (59/61) carried one of the following acquired *bla*_CHDL_ genes: *bla*_OXA-24-like_ (39/59), IS*Aba1*-*bla*_OXA-23-like_ (14/59) or IS*Aba3*-*bla*_OXA-58-like_ (6/59). Whole genome sequence analysis of 15 selected isolates identified the following intrinsic *bla*_OXA-66_ (OXA-51-like; *n* = 15) and acquired class D β-lactamases (CHDLs): IS*Aba1*-*bla*_OXA-23_ (OXA-23-like; *n* = 7), IS*Aba3*-*bla*_OXA-58_-IS*Aba3* (OXA-58-like; *n* = 2) and *bla*_OXA-72_ (OXA-24-like; *n* = 6). The isolates were classified into 21 pulsotypes using PFGE, and the representative 15 isolates were found to belong to sequence type ST2 of the Pasteur MLST scheme from the global IC2 clone. The Oxford MLST scheme revealed the diversity among these studied isolates, and identified five sequence types (ST195, ST208, ST208/ST1806, ST348 and ST425). CHDL-type carbapenemases and insertion elements upstream of the *bla*_CHDL_ genes were found to be widespread among Polish *A. baumannii* clinical isolates, and this contributed to their carbapenem resistance.

## 1. Introduction

In February 2017, the World Health Organization (WHO) published a list of the antibiotic-resistant “priority pathogens” that pose the greatest threat to human health [1]. Carbapenem-resistant *Acinetobacter baumannii* was classified as the most pressing threat in “the critical group”. This non-fermentative, Gram-negative coccobacillus is one of the main causes of severe nosocomial infections, such as ventilator-associated pneumonia, bloodstream infections, bacteremia, urinary tract infections and wound infections, especially in burn patients and post-surgical procedures [2,3]. The increased incidence of carbapenem resistance among clinical isolates, which has risen dramatically over the last twenty years, is of major concern. Resistance to carbapenems in *A. baumannii* is mediated mainly by the production of various carbapenem-hydrolyzing β-lactamases belonging to Ambler classes A (KPC and GES groups), B (IMP, VIM, SIM and NDM groups) and D (OXA-51-like, OXA-23-like, OXA-24-like, OXA-58-like, OXA-143-like and OXA-235-like) [4]. Among the clinical isolates of *A. baumannii,* carbapenem-hydrolyzing class D β-lactamases (CHDLs), also called serine oxacillinases, are the most prevalent. Although they exhibit a weak hydrolysis of carbapenems [3], CHDLs may confer high levels of carbapenem resistance when an insertion sequence (IS) is located upstream of the gene, providing a strong promoter that leads to the overexpression of *bla*_CHDL_ genes [5]. Genes coding for OXA-51-like enzymes, unlike other CHDL enzymes, occur naturally in all *A. baumannii* strains [6].

In this study, the contribution of different carbapenemases to carbapenem resistance among clinical *A. baumannii* strains isolated in one tertiary hospital in Warsaw, Poland, was determined, and 61 imipenem-nonsusceptible isolates were characterized using phenotypic and molecular methods, including genotyping of the isolates. 

## 2. Results

### 2.1. Distribution of bla_CHDL_ Genes and Insertion Sequences

Among all 61 of the collected non-duplicate, imipenem-nonsusceptible clinical isolates of the *Acinetobacter calcoaceticus—A. baumannii* complex, the identification of the *A. baumannii* species was confirmed using *gyrB* multiplex PCR. PCR analysis of the *bla*_CHDL_ genes occurrence among these isolates revealed the presence of intrinsic *bla*_OXA-51-like_ genes and following acquired *bla*_CHDL_ genes: *bla*_OXA-23-like_ in 14 of the 61 (23%) isolates, *bla*_OXA-24-like_ in 39 of the 61 (64%) isolates and *bla*_OXA-58-like_ in 6 of the 61 (10%) isolates. Furthermore, 59 of the 61 isolates harbored no more than two *bla*_CHDL_ genes (*bla*_OXA-51-like_ and one of the acquired *bla*_OXA-23/OXA-24/OXA-58_ genes). The *bla*_OXA-143-like_ genes were not detected in any of the isolates. Two isolates (3%) carried only *bla*_OXA-51-like_ genes, and in these two cases, IS*Aba1* was found upstream of the genes encoding the OXA-51-like enzymes. Each of the *bla*_OXA-23-like_ and *bla*_OXA-58-like_ genes had an upstream IS*Aba1* and IS*Aba3* insertion respectively. Whole genome sequencing was performed on 15 selected *A. baumannii* isolates, including the representative isolates containing each acquired OXA-possessing group, with differing β-lactam sensitivity profiles and CarbAcineto NP assay results, including the time after which positive carbapenemase activity was detected (Appendix A). The whole genome datasets of the 15 strains generated and analyzed during the current study are available at the NCBI BioProject repository (SubmissionID: SUB9082120, BioProject ID: PRJNA701882). Whole genome sequencing (WGS) analysis identified the *bla*_OXA-66_ gene encoding intrinsic OXA-66 belonging to the OXA-51-like family, as well as the *bla*_ADC-30_ gene encoding ADC-30 from the AmpC cephalosporinase family, in all of the isolates. Among the acquired CHDLs and IS elements, the genome analysis revealed the presence of *bla*_OXA-72_ (OXA-24-like family) in six isolates (no. 76, 81, 159, 165, 176 and 195), IS*Aba1* upstream of the *bla*_OXA-23_ gene (OXA-23-like family) in five isolates (no. 96, 113, 118, 129 and 185) and IS*Aba3* upstream and downstream of the *bla*_OXA-58_ (OXA-58-like family) in two isolates (no. 43 and 52;Figure 1). Additionally, in the case of five isolates (no. 86, 87, 96, 129 and 185), the presence of a *bla*_TEM-1_ gene encoding a β-lactamase with a narrow spectrum of activity was demonstrated. 

### 2.2. Antimicrobial Agent Resistance Profiles in Relation to bla_CHDL_ Genes Presence

Table 1 shows the susceptibility pattern of 61 *A. baumannii* isolates for 13 antibacterial agents. The minimal inhibitory concentration (MIC) distribution for the carbapenems, imipenem and meropenem, is shown in Table 2. All 61 *A. baumannii* isolates were determined to be colistin-sensitive, and resistant to ciprofloxacin, piperacillin and piperacillin with tazobactam, regardless of the *bla*_CHDL_ gene carriage. The majority of isolates were resistant to trimethoprim–sulfamethoxazole, with the exception of 5 out of the 39 *bla*_OXA-24-like_-carrying isolates. Of the isolates carrying the acquired *bla*_CHDL_ genes, all isolates containing the IS*Aba1*-*bla*_OXA-23-like_ and *bla*_OXA-24-like_ genes were resistant to both of the carbapenems tested. The highest imipenem and meropenem MIC values were obtained for *bla*_OXA-24-like_carrying isolates (64 mg/L and 128 mg/L, respectively).

### 2.3. Phenotypic Detection of Carbapenemases

Positive carbapenemase activity results using the CarbAcineto NP Test were obtained for the majority of IS*Aba1*-*bla*_OXA-23-like_- (*n* = 12/14) and *bla*_OXA-24-like_-carrying (*n* = 34/39) isolates. For the other *bla*_OXA-23-like_- and *bla*_OXA-24-like_-carrying isolates, as well as all of the isolates containing IS*Aba3*-*bla*_OXA-58-like_ (*n* = 6) and IS*Aba1*-*bla*_OXA-51-like_ genes (*n* = 2), uninterpretable (the optical reading indicated only a slight change in color for the phenol red solution in the test tube) results were observed. In these cases, compared to the internal control (red color tube), after the required incubation time (maximum 2 h), only the development of a red-orange color was observed in the test tube, but not the expected yellow or orange color. The test was performed in duplicate.

Moreover, none of the 61 studied isolates showed MBL-type enzymes based on the results obtained from the double disc diffusion method (DDST) using ethylenediaminetetraacetic acid (EDTA) assay, imipenem and ceftazidime discs.

### 2.4. Molecular Typing and Genome Analysis of the Isolates

To understand how closely related all 61 *A. baumannii* isolates are, they were genotyped using pulsed field gel electrophoresis (PFGE) (Figure 2). Twenty-one pulsotypes (PTs) were determined. The isolates were considered to be a cluster when their similarity was at least 80%. Isolates harboring the *bla*_OXA-24-like_, IS*Aba1*-*bla*_OXA-23-like_ and IS*Aba3-bla*_OXA-58-like_ genes were clustered into 12 PTs, 5 PTs and 3 PTs, respectively. Two isolates carrying only the IS*Aba3-bla*_OXA-51-like_ gene were clustered into two different PTs. Among isolates harboring the *bla*_OXA-24-like_ gene, the majority were isolated in 2012 and 2013 (31/39, 80%) and belonged to 11 different clusters. Most isolates obtained in 2010 and 2011with the *bla*_OXA-24-like_ gene formed a separate cluster (PT A) and for five of these isolates a similarity of over 98% was observed. One isolate from 2012 was also mapped to the PT A pulsotype.

Whole genome analysis of the subsection of isolates (no. 43, 52, 76, 81, 86, 87, 96, 113, 118, 129, 159, 165, 176, 185 and 195) containing the acquired *bla*_CHDL_ genes and from a variety of the pulsotypes revealed that, when multi-locus sequence typing (MLST) typed, they all belonged to the Pasteur sequence type 2 (ST2). The Oxford MLST scheme distinguished five STs among the whole genome sequenced isolates: ST195 (*n* = 3), ST208 (*n* = 1), ST208/ST1806 (*n* = 2), ST348 (*n* = 7) and ST425 (*n* = 2). Information regarding the whole genome sequenced isolates is presented in Table 3. 

## 3. Discussion

An increase in the number of *A. baumannii* strains that are resistant to carbapenems is one of the major therapeutic problems worldwide [3,7,8]. Most of these carbapenem-resistant strains exhibit multidrug-resistance (MDR) or extensively drug-resistance (XDR) profiles. According to the European Centre for Disease Prevention and Control (ECDC) data, more than 50% of invasive *Acinetobacter* spp. strains isolated in Poland in 2015 were defined as carbapenem resistant [9]. Among the 61 carbapenem-nonsusceptible isolates analyzed in our study, all were classified as MDR strains, as defined by Magiorakos et al. [10]. More worryingly, several isolates carrying the genes IS*Aba1*-*bla*_OXA-23-like_ (10 isolates), *bla*_OXA-24-like_ (6 isolates) and IS*Aba1*-*bla*_OXA-51-like_ (1 isolate), showed an XDR resistance profile. An in-depth understanding of the mechanisms causing this widespread resistance is one of the main challenges in the fight to limit the occurrence and spread of carbapenem-resistant strains. 

An analysis of the occurrence of *bla*_CHDL_ genes among the studied isolates revealed the presence of the intrinsic *bla*_OXA-51-like_ gene in all of the isolates. The majority of isolates also carried one of the following acquired *bla*_CHDL_ genes: *bla*_OXA-24-like_, IS*Aba1*-*bla*_OXA-23-like_ or IS*Aba3*-*bla*_OXA-58-like_. In our study, the IS*Aba1* sequence was also detected upstream of *bla*_OXA-51-like_ genes in two isolates in which no acquired genes encoding the remaining CHDL enzymes were found. The presence of an insertion sequence located upstream of *bla*_CHDL_ leads to an overexpression of this gene, and contributes to the resistance of *A. baumannii* strains to β-lactams, including carbapenems [5]. So far, the occurrence of the following IS elements IS*Aba1*, IS*Aba2*, IS*Aba3*, IS*Aba4*, IS*Aba9*, IS*Aba10*, IS*18* and IS*Aba825*, associated with carbapenem resistance in *A. baumannii* clinical strains, has been described [11,12,13,14]. IS*Aba1* is the most common insertion sequence identified in *A. baumannii,* as well as in other *Acinetobacter* species, worldwide [12,14]. In clinical *A. baumannii* strains, the IS*Aba1* element was found primarily upstream of the *bla*_OXA-23-like_ genes and, less often, upstream of the *bla*_OXA-51-like_ and *bla*_OXA-58-like_ genes. The group of OXA-23-like enzymes is the most commonly acquired resistance CHDL carbapenemase worldwide [3,15,16,17,18,19]. In our study, all of the isolates with the *bla*_OXA-23-like_ genes possessed IS*Aba1* upstream. This is the first data concerning the frequency of IS*Aba1* upstream of *bla*_OXA-23-like_ genes in *A. baumannii* isolates from Poland. WGS analysis confirmed the presence of IS*Aba1*-*bla*_OXA-23_ gene sequences. So far, in isolates studied worldwide, the presence of other insertion sequences such as IS*Aba4* and IS*Aba10* upstream of *bla*_OXA-23-like_ genes has been described [12]. Interestingly, among the strains examined in this study, more were carrying the *bla*_OXA-24-like_ gene than the IS*Aba1*-*bla*_OXA-23-like_ gene (64% versus 23%, respectively). A few studies on *A. baumannii* strains producing OXA-24 in Poland also indicated a high frequency of occurrence among strains isolated from hospitals. Chmielarczyk et al. reported that 79% of the isolates they studied possessed the *bla*_OXA-24_ gene, whereas in the study by Nowak et al., only 49% of the isolates studied possessed the *bla*_OXA-24_ gene [20,21]. Reports from Poland appear to be contrary to the data from other countries. OXA-24-like carbapenemases produced by *A. baumannii* have been found in many European (Bulgaria, France, Portugal, Spain and Sweden) and non-European (Israel, United States and Colombia) countries; however, in most reports, OXA-24 gave way to OXA-23, making the latter more frequently occurring [15,17,22,23,24,25,26]. An insertion sequence has never been found upstream of *bla*_OXA-24-like_ genes. The genome analysis revealed the presence of the *bla*_OXA-72_ gene (from *bla*_OXA-24-like_ family) in six isolates analyzed in this study. This variant retains its carbapenemase activity. According to Liu et al., imipenem MIC values for isolates producing this enzyme is at least 64 mg/L, which is concordant with our results [17]. A third group of the acquired *bla*_CHDL_ genes identified among the Polish isolates tested was IS*Aba3*-*bla*_OXA-58-like_. The *bla*_OXA-58_ gene is currently being reported worldwide, and its increased expression is always associated with IS elements upstream of this gene. In *A. baumannii* strains isolated from different European countries, the following insertion elements have been found in the *bla*_OXA-58-like_ gene promoters: IS*Aba1*, IS*Aba2*, IS*Aba3,* IS*Aba3*-_like_, IS*18* and IS*Aba825* [8,13,27,28]. To date, there are only sparse data on the occurrence of *bla*_OXA-58-like_ genes in *A. baumannii* strains isolated in Poland, but none contained information about the presence of IS elements upstream of these genes [15,29]. This is the first time the IS*Aba3* sequence has been detected upstream of the *bla*_OXA-58-like_ genes in *A. baumannii* isolates from Poland, as well as in all six of the OXA-58-producing isolates. Moreover, WGS analysis showed the presence of IS*Aba3* elements both upstream and downstream of the *bla*_OXA-58_ gene in the two isolates tested. It has been reported that often the *bla*_OXA-58-like_ genes in *A. baumannii* strains are flanked on two sides with the same or different IS elements [4,7,12].

Overall, using PFGE for genotyping all 61 isolates collected over 2010-2014 did not show any long-term epidemiological incidents caused by any carbapenem resistant *A. baumannii* strains among patients in the tested hospital. Only a few isolates showed a correlation with a 95% similarity, indicating the transmission of strains between several patients (among two, three, and five patients). The largest number (*n* = 4) of identical (100% similarity in PFGE) isolates (OXA-24-producing) from different patients was received in 2010. The MLST results revealed that all whole genome sequenced isolates belonged to ST2 (by the Pasteur scheme) which is a highly predominant clone in Poland, as well as in other European countries [15,29,30]. In Poland, carbapenem-resistant strains of *A. baumannii* belonging to other clones, such as ST1, ST5 and ST193, have rarely been isolated [29]. Analysis of the other four out of seven house-keeping genes in the Oxford scheme typing in relation to the Pasteur scheme allows for the differentiation of strains, especially within ST2, that dominate in Europe. The Oxford scheme revealed diversity among studied isolates and distinguished five sequence types (ST195, ST208, ST208/ST1806, ST348 and ST425). So far, genetic typing of Polish strains of *A. baumannii* has not been carried out using the Oxford scheme. Among the *A. baumannii* strains isolated in Europe, the following above-mentioned sequence types have been described: ST195 in Croatia, ST208 in Bulgaria and Greece, ST348 in Germany and ST425 in Greece [31,32,33,34,35]. Carbapenem-resistant *A. baumannii* ST195 and ST208 are widespread in the world, and are dominant mainly in the Far East, e.g., in China [36,37,38,39,40]. Furthermore, the ST208/ST1806 strains have been isolated in South Korea [41].

In our study, the majority of isolates carrying the *bla*_OXA-24-like_ and IS*Aba1*-*bla*_OXA-23-like_ genes demonstrated the highest carbapenem MIC values, and were the only isolates that presented a positive CarbAcineto NP test result. According to Dortet et al., this test is capable of detecting the production of various CHDLs, such as OXA-23, OXA-24, OXA-58 and OXA-143 groups, by *A. baumannii* strains with a high level of resistance to carbapenems, i.e., MICs 8 - >32 mg/L [42]. Positive carbapenemase activity results in the CarbAcineto NP Test were obtained for the majority of the IS*Aba1*-*bla*_OXA-23-like_ (*n* = 12/14) and *bla*_OXA-24-like_-carrying (*n* = 34/39) isolates. In this study, the results of the CarbAcineto NP test for several *bla*_OXA-23-like_- and *bla*_OXA-24-like_-carrying isolates (2/14 and 5/39, respectively), as well as for all isolates with the IS*Aba3*-*bla*_OXA-58-like_ (*n* = 6) and IS*Aba1*-*bla*_OXA-51-like_ genes (*n* = 2), were uninterpretable. Even though the CarbAcineto NP test is able to detect *A. baumannii* CHDL-producing isolates, it is still possible that the results would be ambiguous. Literacka et al., in their study, demonstrated that for one OXA-24-like, three OXA-23-like and two OXA-58-like-producing *A. baumannii* isolates, the results of the CarbAcineto NP Test were uninterpretable [43]. In such cases, expression levels of the genes encoding CHDL enzymes may be low or inhibited, which would suggest that some other resistance mechanism affecting carbapenem activity dominates in those isolates, giving high carbapenem MIC values. Perhaps various resistance mechanisms interact simultaneously in these few isolates, such as efflux pumps, changes in outer membrane proteins and, to a lesser extent, carbapenemases. 

## 4. Materials and Methods

### 4.1. Bacterial Strains

Here, 61 non-repetitive imipenem-nonsusceptible *A. baumannii* isolates were collected in 2009 (*n* = 2), 2010 (*n* = 9), 2011 (*n* = 4), 2012 (*n* = 18), 2013 (*n* = 27) and 2014 (*n* = 1) from patients hospitalized in one tertiary hospital in Warsaw, Poland. The isolates were recovered from various clinical specimens, including respiratory tract samples (15, 24.6%), wound swabs (15, 24.6%), urine (15, 24.6%), blood (3, 4.9%) and other specimens (13, 21.3%). The isolates were initially identified using the VITEK 2 system (bioMérieux, Mercy l’Etoile, France) and were assigned to the *Acinetobacter calcoaceticus–A. baumannii* complex. Identification of *A. baumannii* was confirmed by *gyrB* multiplex PCR, as previously described [44,45]. 

### 4.2. Antimicrobial Susceptibility Testing

The isolates were first determined to be imipenem-nonsusceptible using a disk diffusion method utilizing an imipenem disk (10 μg, Becton, Dickinson and Company, Franklin Lakes, NJ, USA), according to Clinical and Laboratory Standards Institute (CLSI) recommendations (2018) [46]. Minimal inhibitory concentrations (MICs) of piperacillin, piperacillin/tazobactam, ceftazidime, cefepime, imipenem, meropenem, gentamicin, tobramycin, ciprofloxacin, levofloxacin, colistin and trimethoprim/sulfamethoxazole were determined using the VITEK 2 system (bioMérieux, Mercy l’Etoile, France) and relevant antimicrobial susceptibility testing cards. 

The MICs of imipenem and meropenem were also determined with the microdilution method in a Mueller Hinton II (MH II) broth medium (Becton, Dickinson and Company, Franklin Lakes, NJ, USA), according to CLSI guidelines (2012, 2018) [46,47]. *Escherichia coli* ATCC 25,922 was used as the reference strain for quality control. 

### 4.3. Phenotypic Detection of Carbapenemase Production

#### 4.3.1. Detection of Metallo-β-Lactamases (MBL)

The MBL mechanism of resistance was detected using the double disc diffusion method (DDST) [48]. Disks of 10 μg of imipenem (Becton, Dickinson and Company, Franklin Lakes, NJ, USA) and 30 μg of ceftazidime (Becton, Dickinson and Company, Franklin Lakes, NJ, USA) were placed at a distance of 20 mm (center to center) from a disk containing 0.01 mL of 0.5 mM EDTA solution. The presence of an extended inhibition zone of antibiotics towards the disc with EDTA was interpreted as a positive result for metallo-β-lactamase-producing strains screening. 

#### 4.3.2. CarbAcineto NP Test

The CarbAcineto NP test was adapted from the updated version of the Carba NP test, used for the detection of carbapenemase-producing *Enterobacteriaceae* and *Pseudomonas* spp. The test was performed as described by Dortet et al. [42].

### 4.4. Molecular Detection of bla_OXA-like_ Genes and ISs Upstream of bla_CHDL_ Genes

Genomic DNA was extracted using the Genomic Mini isolation kit (A&A Biotechnology, Gdynia, Poland), according to manufacturer’s instructions. Detection of the *bla*_CHDL_ genes was performed using (1) duplex PCR for *bla*_OXA-51-like_ and *bla*_OXA-23-like_, (2) duplex PCR for *bla*_OXA-24-like_ and *bla*_OXA-58-like_ and (3) singleplex for *bla*_OXA-143-like_, and (4) singleplex was used for detection of the insertion sequences (ISs) upstream of *bla*_OXA-51-like_, *bla*_OXA-23-like_, *bla*_OXA-24-like_ and *bla*_OXA-58-like_. The primers used are listed in Table 4. The PCR reactions were performed using Hypernova polymerase (Blirt S.A., Gdańsk, Poland) with the following amplification parameters: (1) all duplex PCRs were performed at 95 °C for 4 min, followed by 25 cycles of 30 s at 95 °C, 30 s at 55 °C, 50 s at 72 °C and a final extension of 3 min at 72 °C, and (2) singleplex PCR for *bla*_OXA-143-like_ was performed at 95 °C for 4 min, followed by 25 cycles of 30 s at 95 °C, 30 s at 54 °C, 45 s at 72 °C and a final extension of 3 min at 72 °C. In addition, (3) singleplex PCRs for the ISs upstream each *bla*_CHDL_ were performed at 95 °C for 4 min, followed by 25 cycles of 30 s at 95 °C, 30 s at 56 °C, 75 s at 72 °C and a final extension of 3 min at 72 °C.

### 4.5. Pulsed Field Gel Electrophoresis (PFGE)

All 61 *A. baumannii* isolates were typed by PFGE according to a previous protocol [50] with modifications. An overnight culture of bacteria was suspended in 150 μL of a cell suspension buffer and was mixed with 20 μL of 20 mg/mL Proteinase K (Promega GmbH, Walldorf, Germany) and 170 μL of 1.5% low melting agarose (SeaKem Gold Agarose (Lonza, Basel, Switzerland)) and distributed in a plug mold. The genomic DNA in agarose plugs was lysed in 2.5 mL of cell lysis solution supplemented with 20 μL of 20 mg/mL Proteinase K (Promega GmbH, Walldorf, Germany) and 7.5 μL of 10 mg/mL RNase (Sigma, St. Louis, MO, USA), and was washed and digested with 20 U *ApaI* restriction enzyme (ABO, Gdańsk, Poland). Electrophoresis was performed using the CHEF DR II system (Bio-Rad, Hercules, CA, USA). The migration conditions were as follows: switch angle 120, voltage 6 V/cm, temperature 14 °C and a two-blocks program with a total run time of 22 h (first-block 1-8 s for 15 h, and second-block 5–20 s for an additional 7 h). The genomic DNA of *Salmonella* serotype Braenderup strain (H9812) digested with *XbaI* (ABO, Gdańsk, Poland) and Lambda-DNA Ladder PFG Marker (New England BioLabs, Ipswich, MA, USA) were used as the DNA molecular-weight markers [51]. The PFGE patterns were analyzed using GelCompar II software (Applied Maths, Sint-Martens-Latem, Belgium) with the Dice coefficient and clustering by unweighted pair group method with arithmetic mean (UPGMA) with 1% tolerance. The isolates were clustered in the PFGE pulsotypes (PTs) in accordance with the recommendations of Tenover et al. [52]. 

### 4.6. Whole Genome Sequencing (WGS) and Multi-Locus Sequence Typing (MLST)

Several *A. baumannii* isolate genomes were whole genome sequenced. This was performed using Public Health England - Genomic Services and Development Unit (PHE-GSDU) on the HiSeq 2500 System (Illumina, CambridgeZ, UK) with paired end read lengths of 150 bp. A minimum of 150 Mb of Q30 quality data were obtained for each isolate. FastQ files were quality trimmed using Trimmomatic 0.32 [53]. SPAdes 3.1.1 was used to produce draft chromosomal assemblies, and contigs less than 500 bp were filtered out [54]. All of the above sequencing analyses were performed using PHE Galaxy [55]. Using the WGS data, the sequence types (STs) were determined using both the Pasteur and Oxford MLST databases [56]. In the case of the Pasteur MLST scheme, the *cpn60, fusA, gltA, pyrG, recA, rplB* and *rpoB* gene sequences were analyzed*,* while the Oxford MLST scheme analysis included the *gltA*, *gyrB*, *gdhB*, *recA*, *cpn60*, *gpi* and *rpoD* gene sequences.

## 5. Conclusions

From the *A. baumannii* clinical strains isolated in Poland, the dominating carbapenem resistance mechanism was CHDL enzymes, mainly those from the OXA-24-like family. The presence of insertion sequences flanking *bla*_OXA-58-like_ (IS*Aba3*) and *bla*_OXA-23-like_ (IS*Aba1*) genes, which enhance the gene expression, was demonstrated. All tested isolates belong to sequence type ST2 (of the Pasteur MLST scheme) from the global IC2 clone. The Oxford MLST scheme allowed for the differentiation of the sequence types of these isolates.

## Figures and Tables

**Figure 1 ijms-22-02486-f001:**
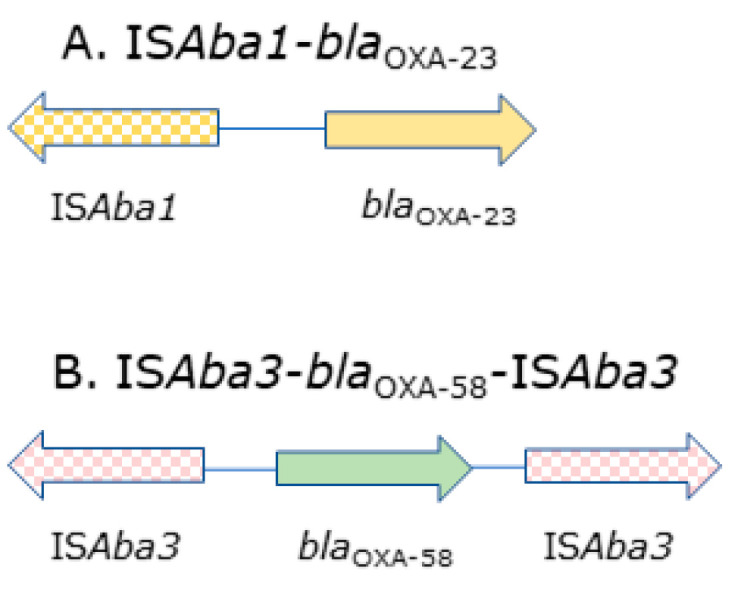
The intrinsic *bla*_CHDL_ genes with insertion sequence (IS) element clusters found in the following clinical isolates: (**A**) no. 86, 87, 96, 113, 118, 129 and 185; (**B**) no. 43 and 52.

**Figure 2 ijms-22-02486-f002:**
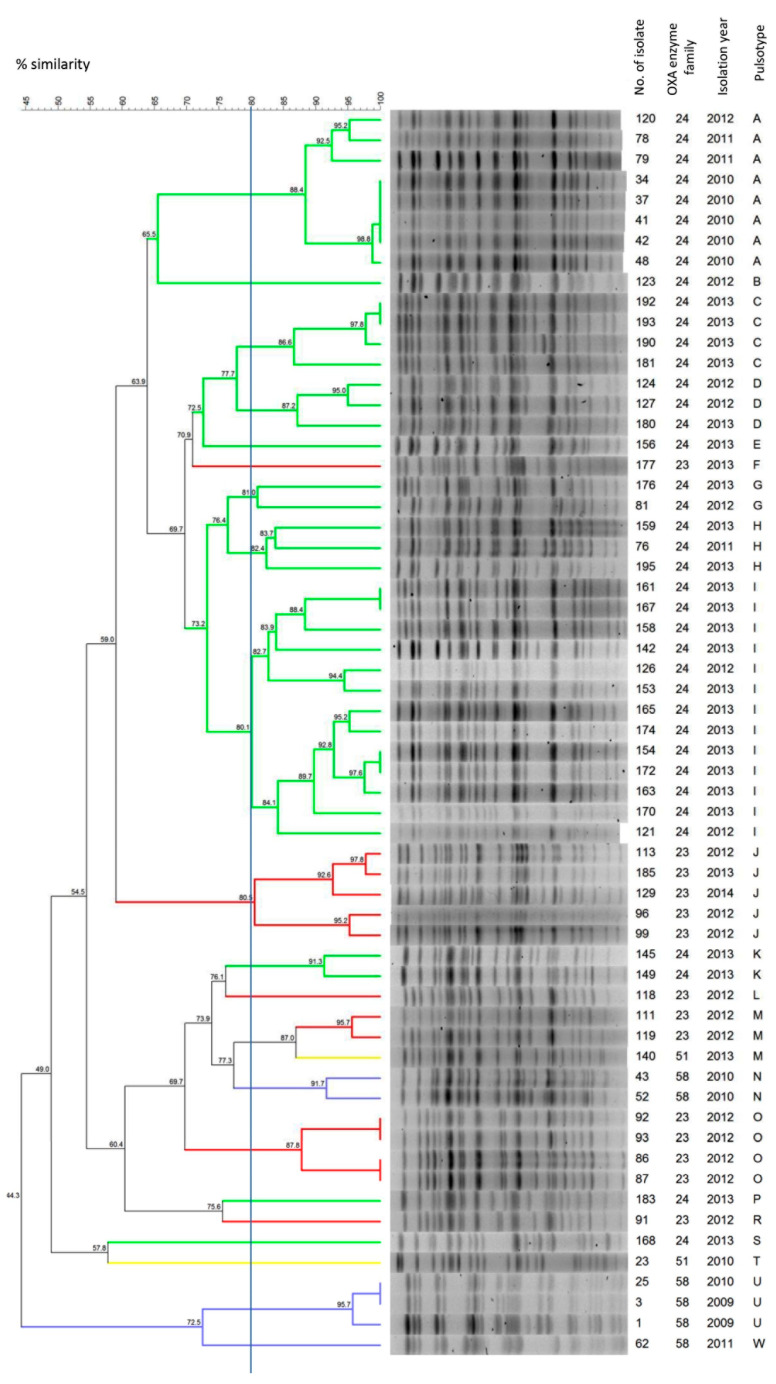
Analysis of pulsed field gel electrophoresis (PFGE) patterns. The dendrogram presents PFGE profiles’ percentage similarity, the presence of the OXA enzyme family and the isolation year of the *A*. *baumannii* clinical isolates. The solid line indicates 80% similarity and is used to define the pulsotypes. Isolates carrying different groups of the acquired *bla*_CHDL_ genes were marked with the following colors: red, *bla*_OXA-23-like_; green, *bla*_OXA-24-like_; and blue, *bla*_OXA-58-like_. Isolates harboring only intrinsic *bla*_OXA-58-like_ gene were marked in yellow.

**Table 1 ijms-22-02486-t001:** The susceptibility profiles of *A. baumannii* isolates (*n* = 61) carrying *bla*_CHDL_ genes.

Antimicrobial Agent	No. of Isolates with the Indicated Susceptibility Carrying *bla*_CHDL_ Genes											
	IS*Aba1*- *bla*_OXA-51-like_ (*n* = 2)			*bla* _OXA-51-like_								
				IS*Aba3*- *bla*_OXA-58-like_ (*n* = 6)			IS*Aba1*-*bla*_OXA-23-like_ (*n* = 14)			*bla*_OXA-24-like_ (*n* = 39)		
	R	I	S	R	I	S	R	I	S	R	I	S
Ampicillin/sulbactam	1	0	1	0	1	5	3	8	3	3	4	32
Cefepime	0	1	1	0	4	2	12	2	0	10	28	1
Ceftazidime	2	0	0	2	4	0	14	0	0	39	0	0
Imipenem	1	1	0	6	0	0	14	0	0	39	0	0
Meropenem	2	0	0	0	6	0	14	0	0	39	0	0
Piperacillin	2	0	0	6	0	0	14	0	0	39	0	0
Piperacillin/tazobactam	2	0	0	6	0	0	14	0	0	39	0	0
Ciprofloxacin	2	0	0	6	0	0	14	0	0	39	0	0
Levofloxacin	1	1	0	2	4	0	14	0	0	16	23	0
Gentamicin	2	0	0	2	0	4	8	0	6	34	3	2
Tobramicin	1	0	1	2	0	4	11	0	3	32	5	2
Colistin	0	0	2	0	0	6	0	0	14	0	0	39
Trimethoprim/Sulfamethoxazole	2	0	0	6	0	0	14	0	0	34	0	5

R—resistant; I—intermediate; S—susceptible.

**Table 2 ijms-22-02486-t002:** The carbapenem minimal inhibitory concentration (MIC) distribution of the *bla*_CHDL_-carrying *A. baumannii* isolates (*n* = 61).

Groups of Isolates Carrying the following Genes		Carbapenem	No. of Isolates with the Indicated MIC Values					
			4 mg/L	8 mg/L	16 mg/L	32 mg/L	64 mg/L	128 mg/L
IS*Aba1*-*bla*_OXA-51-like_ (*n* = 2)		Imipenem	1	1	0	0	0	0
		Meropenem	0	1	1	0	0	0
*bla* _OXA-51-like_	IS*Aba3*-*bla*_OXA-58-like_ (*n* = 6)	Imipenem	0	2	4	0	0	0
		Meropenem	0	5	0	0	1	0
	IS*Aba1*-*bla*_OXA-23-like_ (*n* = 14)	Imipenem	0	1	9	4	0	0
		Meropenem	0	2	3	4	5	0
	*bla*_OXA-24-like_ (*n* = 39)	Imipenem	0	2	14	12	11	0
		Meropenem	0	0	0	2	17	20

**Table 3 ijms-22-02486-t003:** Sequence types, β-lactamases and epidemiological data of the whole genome sequenced *A. baumannii* isolates (*n* = 15).

Oxford ST *	Acquired CHDLs Enzyme (Family)	Other β-lactamases	Isolate Number	Isolation Year	Clinical Material
195	OXA-23 (OXA-23-like)	ADC-30, TEM-1	96	2012	Respiratory tract sample
195	OXA-23 (OXA-23-like)	ADC-30, TEM-1	185	2013	Urine
195	OXA-23 (OXA-23-like)	ADC-30, TEM-1	129	2014	Urine
208	OXA-72 (OXA-24-like)	ADC-30	176	2013	Urine
208/1806	OXA-58 (OXA-58-like)	ADC-30	43	2010	Fistula
208/1806	OXA-58 (OXA-58-like)	ADC-30	52	2010	Respiratory tract sample
348	OXA-72 (OXA-24-like)	ADC-30	76	2011	Wound
348	OXA-72 (OXA-24-like)	ADC-30	81	2012	Wound
348	OXA-72 (OXA-24-like)	ADC-30	159	2013	Urine
348	OXA-72 (OXA-24-like)	ADC-30	165	2013	Respiratory tract sample
348	OXA-72 (OXA-24-like)	ADC-30	195	2013	Wound
348	OXA-23 (OXA-23-like)	ADC-30	113	2012	Urine
348	OXA-23 (OXA-23-like)	ADC-30	118	2012	Rectal swab
425	OXA-23 (OXA-23-like)	ADC-30, TEM-1	86	2012	Urine
425	OXA-23 (OXA-23-like)	ADC-30, TEM-1	87	2012	Urine

* All 15 studied isolates are distinguished into five sequence types by the Oxford multi-locus sequence typing (MLST) scheme, and belong to ST2 of the Pasteur MLST scheme.

**Table 4 ijms-22-02486-t004:** Primers used for amplification.

Target Genes	Primer	Sequence (5’ → 3’)	Product Length (bp)	Reference
*bla* _OXA-51-like_	51/F	taatgctttgatcggccttg	352	[49]
	51/R	tggattgcacttcatcttgg		
*bla* _IS*Aba1*+OXA-51-like_	ISAba1/F	aatcacaagcatgatgagcg	880	this study
	51/R	tggattgcacttcatcttgg		[49]
*bla* _OXA-58-like_	58/F	aagtattggggcttgtgctg	598	[49]
	58/R	cctctgcgctctacatac		this study
*bla* _IS*Aba3*+OXA-58-like_	ISAba3/F	aggcaggttggacatttgat	755	this study
	58/R	cctctgcgctctacatac		this study
*bla* _OXA-23-like_	23/F	gatcggattggagaaccaga	501	[49]
	23/R	catttctgaccgcatttccat		this study
*bla* _IS*Aba1*+OXA-23-like_	ISAba1/F	aatcacaagcatgatgagcg	962	this study
	23/R	catttctgaccgcatttccat		this study
*bla* _OXA-24-like_	24/F	ggttagttggcccccttaaa	248	[49]
	24/R	agttgagcgaaaaggggatt		
*bla* _IS*Aba1*+OXA-24-like_	ISAba1/F	aatcacaagcatgatgagcg	448	this study
	24/R	agttgagcgaaaaggggatt		[49]
*bla* _OXA-143-like_	143/F	cagtgcatgctcatctattc	460	this study
	143/R	ggccaaccaaccagaagtt		this study

## Data Availability

The whole genome datasets of the 15 strains are available at the NCBI BioProject repository (SubmissionID: SUB9082120, BioProject ID: PRJNA701882).

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
