# Peer review of "Analysis of blaCHDL Genes and Insertion Sequences Related to Carbapenem Resistance in Acinetobacter baumannii Clinical Strains Isolated in Warsaw, Poland"

_ijms, 2021, doi:10.3390/ijms22052486_

Round 1

Reviewer 1 Report

In their manuscript entitled “Analysis of blaCHDL genes and insertion sequences related to carbapenem resistance in Acinetobacter baumannii clinical strains isolated in Warsaw, Poland”, SĹ‚oczyĹ„ska and co-authors describe the characterization of 61 imipenem-nonsusceptible A. baumannii isolates obtained in Warsow, Poland, by phenotypic and molecular methods. PFGE and MLST were utilized for genotyping. Phenotypic characterization of the isolates was performed by drug-susceptibility profiling based on disk diffusion assays and MIC values determination, as well as by the CarbaAcinetoNP test to detect carbapenemase activity. The production of metallo-β-lactamases by the 61 isolates was also investigated. A subset of 15 isolates were further characterized by whole enome sequencing. The manuscript is in general well organized and written, on an important topic that is the increase of carbapenem-resistant A. baumannii strains worldwide.

There are however some missing information that the authors could add to the work to increase its quality.  For instance, the authors mention and show some data based on the whole genome sequence of 15 selected isolates, but there is no mention to a repository of the data which I suppose is not available. The availability of data would support the authors conclusions, allowing the reader to confirm the statements of the authors. Another issue is related to the pulsotypes. Data shown is a final dendrogram with the percentage similarities. I suggest that the authors add a photograph of the gel after electrophoresis with the profile for each isolate after the pulsotype, something like a strip to show the results. Another difficult to understand is the rationale for the selection of 15 isolates for WGS. Why these 15? And what is the significance of performing PCR amplifications of genes and then sequence the whole genomes. Isn´t is redundant? The authors claim that MLST was performed using two different sets of housekeeping genes, the Oxford and the Pasteur schemes. Since the Pasteur scheme was unable to distinguish the isolates and all were found to belong to the same Pasteur ST2, is it worth to use a column with the Pasteur ST for the isolates in Table 3? A line on the legend stating this might be enough. In the Materials and Methods section I missed the preparation of samples for PFGE analysis. How were cells collected for DNA extraction? Were the cells immobilized in agarose plugs prior DNA restriction? This information is missing and one reads the methods and thinks that sample preparation is similar to a normal DNA extraction, restriction and restriction analysis after gel electrophoresis.

Author Response

The answers for the Reviewer 1.

There are however some missing information that the authors could add to the work to increase its quality.  For instance, the authors mention and show some data based on the whole genome sequence of 15 selected isolates, but there is no mention to a repository of the data which I suppose is not available. The availability of data would support the authors conclusions, allowing the reader to confirm the statements of the authors.

As suggested by the Reviewer, the whole genome sequences of the studied 15 strains were submitted to NCBI GenBank.  In section Results we added the sentence:

The whole genome datasets of 15 strains generated and analyzed during the current study are available at the NCBI BioProject repository (SubmissionID: SUB9082120, BioProject ID: PRJNA701882).

Another issue is related to the pulsotypes. Data shown is a final dendrogram with the percentage similarities. I suggest that the authors add a photograph of the gel after electrophoresis with the profile for each isolate after the pulsotype, something like a strip to show the results.

As suggested by the Reviewer, the Figure 2 has been changed. We have added photographs of the PFGE gels to the dendrogram presenting PFGE profiles.

Another difficult to understand is the rationale for the selection of 15 isolates for WGS. Why these 15? And what is the significance of performing PCR amplifications of genes and then sequence the whole genomes. Isn´t is redundant?

The PCR analysis of occurrence of blaCHDL genes, both intrinsic and acquired, and the insertion sequences upstream of blaCHDL genes allows only to reveal the presence of: (1) a gene family, not a specific gene, and (2) an IS sequence without the possibility of analyzing its nucleotide sequence in detail and studying the possible presence of other ISs upstream and downstream of the blaCHDL genes. The PCR is a very good accurate method of screening a group of a large number of strains. However, sequencing must be performed for in-depth analysis of blaCHDL genes and the surrounding nucleotide sequences. The use of WGS made it possible not only to determine the blaCHDL genes and the IS sequences surrounding them, but also to analyze the genomes in the search for genes encoding other beta-lactamases, including other carbapenemases, and to determine the sequence types (ST) according to the Pasteur and Oxford MLST scheme. WGS analysis provides a lot of data that can be analyzed in many directions.

To explain the choice of 15 isolates for WGS analysis, Table S1 in the supplementary information was added to the manuscript. Moreover, in section Results we changed the sentence:

15 selected A. baumannii  isolates, including representative isolates containing each acquired OXA-possessing group, were whole genome sequenced.

new sentence:

15 selected A. baumannii  isolates, including representative isolates containing each acquired OXA-possessing group, with variety beta-lactam sensitivity profiles and CarbAcinetoNP assay results, including the time after which the carbapenemase activity were positive (Table S1), were whole genome sequenced.

The authors claim that MLST was performed using two different sets of housekeeping genes, the Oxford and the Pasteur schemes. Since the Pasteur scheme was unable to distinguish the isolates and all were found to belong to the same Pasteur ST2, is it worth to use a column with the Pasteur ST for the isolates in Table 3? A line on the legend stating this might be enough.

As suggested by the Reviewer, in Table 3 the Pasteur scheme performance column has been removed and the information has been added to the legend below the table.

In the Materials and Methods section I missed the preparation of samples for PFGE analysis. How were cells collected for DNA extraction? Were the cells immobilized in agarose plugs prior DNA restriction? This information is missing and one reads the methods and thinks that sample preparation is similar to a normal DNA extraction, restriction and restriction analysis after gel electrophoresis.

As suggested by the Reviewer, in section Materials and Methods we added the following text:

All 61 A. baumannii  isolates were typed by PFGE according to a previous protocol [50] with modifications. An overnight culture of bacteria was suspended in 150 μL of cell suspension buffer and was mixed with 20 μL of 20 mg/mL Proteinase K (Promega) and 170 μL of 1.5% low melting agarose - SeaKem Gold Agarose (Lonza), and distributed in a plug mold. Genomic DNA in agarose plugs was lysed in 2.5 mL of cell lysis solution supplemented with 20 μL of 20 mg/mL Proteinase K (Promega) and 7.5 μL of 10 mg/mL RNase (Sigma), washed and digested with 20 U ApaI restriction enzyme (Thermo Scientific).

Reviewer 2 Report

The manuscript entilted as Analysis of blaCHDL genes and insertion sequences related to carbapenem resistance in Acinetobacter baumannii clinical strains isolated in Warsaw, Poland. 

The topic is very significant as drug-resistant A. baumannii infections are very concerning. There are some minor editing are required before publication. 

Author Response

The answers for the Reviewer 2.

The section Abstract, line 27.

The word “method” was added - PFGE and MLST methods.

We propose to leave the abbreviations of the words PFGE and MLST. Abstract contains 200 words which is the limitable number according to the instructions for the author.

The section Introduction, line 63.

The word “imipenem-unsusceptible” has been changed to “imipenem-nonsusceptible”

The section Results:

- line 73

„Single isolates harbored no more than two blaCHDL genes (blaOXA-51-like and one of the acquired blaOXA-23/OXA-24/OXA-58 genes).”

It has been changed to: “Furthermore 59 out of 61 isolates harbored no more than two blaCHDL genes (blaOXA-51-like and one of the acquired blaOXA-23/OXA-24/OXA-58 genes).”

- line 78

To explain the choice of 15 isolates for WGS analysis, Table S1 in the supplementary information was added to the manuscript. Moreover, in section Results we changed the sentence:

15 selected A. baumannii  isolates, including representative isolates containing each acquired OXA-possessing group, were whole genome sequenced.

new sentence:

15 selected A. baumannii  isolates, including representative isolates containing each acquired OXA-possessing group, with variety beta-lactam sensitivity profiles and CarbAcinetoNP assay results, including the time after which the carbapenemase activity were positive (Table S1), were whole genome sequenced.

  • line 300

we changed “manufacture instructions” to “ manufacturer’s instructions”

Following the Reviewer’s recommendations, minor editing errors such as multiple spaces between words and “mg/l” was changed to “mg/L”, were corrected throughout the manuscript.

Round 2

Reviewer 1 Report

The manuscript was adequately revised taking into consideraion the criticisms raised. Therefore, I recommend the acceptance of the manuscript for publication.